# DNAH3 deficiency causes flagellar inner dynein arm loss and male infertility in humans and mice

Xiang Wang[1,2†], Gan Shen[1†], Yihong Yang[3†], Chuan Jiang[1†], Tiechao Ruan[4†], Xue Yang[1], Liangchai Zhuo[1], Yingteng Zhang[1], Yangdi Ou[5], Xinya Zhao[6], Shunhua Long[7,8], Xiangrong Tang[7,8], Tingting Lin[7,8*], Ying Shen[1,2*]

[1]Department of Obstetrics/Gynecology, Key Laboratory of Obstetric, Gynecologic and Pediatric Diseases and Birth Defects of Ministry of Education, West China Second University Hospital, Sichuan University, Chengdu, China; [2]NHC Key Laboratory of Chronobiology, Sichuan University, Chengdu, China; [3]Reproduction Medical Center of West China Second University Hospital, Key Laboratory of Obstetric, Gynecologic and Pediatric Diseases and Birth Defects of Ministry of Education, Sichuan University, Chengdu, China; [4]Department of Pediatrics, West China Second University Hospital, Sichuan University, Chengdu, China; [5]West China School of Medicine, Sichuan University, Chengdu, China; [6]West China School of Basic Medicine and Forensic Medicine, Sichuan University, Chengdu, China; [7]Chongqing Key Laboratory of Human Embryo Engineering, Center for Reproductive Medicine, Women and Children's Hospital of Chongqing Medical University, Chongqing, China; [8]Chongqing Clinical Research Center for Reproductive Medicine, Chongqing Health Center for Women and Children, Chongqing, China

*For correspondence:
yuting9263@163.com (TL);
yingcaishen01@163.com (YS)

[†]These authors contributed equally to this work

Competing interest: The authors declare that no competing interests exist.

## eLife Assessment

This **important** study identifies biallelic variants of DNAH3 in unrelated infertile men and reports infertility in DNAH3 knockout mice. The authors demonstrate that compromised DNAH3 activity decreases the expression of IDA-associated proteins in the spermatozoa of human patients and knockout mice, providing **convincing** evidence that DNAH3 is a novel pathogenic gene for asthenoteratozoospermia and male infertility. The study will be of substantial interest to clinicians, reproductive counselors, embryologists, and basic researchers working on infertility and assisted reproductive technology.

**Abstract** Axonemal protein complexes, including the outer and inner dynein arms (ODA/IDA), are highly ordered structures of the sperm flagella that drive sperm motility. Deficiencies in several axonemal proteins have been associated with male infertility, which is characterized by asthenozoospermia or asthenoteratozoospermia. Dynein axonemal heavy chain 3 (DNAH3) resides in the IDA and is highly expressed in the testis. However, the relationship between DNAH3 and male infertility is still unclear. Herein, we identified biallelic variants of *DNAH3* in four unrelated Han Chinese infertile men with asthenoteratozoospermia through whole-exome sequencing (WES). These variants contributed to deficient DNAH3 expression in the patients' sperm flagella. Importantly, the patients represented the anomalous sperm flagellar morphology, and the flagellar ultrastructure was severely disrupted. Intriguingly, *Dnah3* knockout (KO) male mice were also infertile, especially showing the severe reduction in sperm movement with the abnormal IDA and mitochondrion structure. Mechanically, nonfunctional DNAH3 expression resulted in decreased expression of IDA-associated proteins

in the spermatozoa flagella of patients and KO mice, including DNAH1, DNAH6, and DNALI1, the deletion of which has been involved in disruption of sperm motility. Moreover, the infertility of patients with *DNAH3* variants and *Dnah3* KO mice could be rescued by intracytoplasmic sperm injection (ICSI) treatment. Our findings indicated that *DNAH3* is a novel pathogenic gene for asthenoteratozoospermia and may further contribute to the diagnosis, genetic counseling, and prognosis of male infertility.

## Introduction

Infertility is a global public health and social problem that affects approximately one in six couples worldwide (*Cox et al., 2022*). Male infertility, which accounts for half of infertile cases, is a multifactorial disease with common phenotypes, including oligo/azoospermia (poor sperm count or absence of spermatozoa); teratozoospermia (aberrant sperm morphology); asthenozoospermia (weakened sperm motility); and a combination of these phenotypes, such as asthenoteratozoospermia, oligoasthenozoospermia, oligoteratozoospermia, and oligoasthenoteratozoospermia (*Eisenberg et al., 2023*; *Agarwal et al., 2015*).

Asthenoteratozoospermia is one of the most common phenotypes of male infertility, and genetic factors have been established as the predominant cause of asthenoteratozoospermia. Multiple morphological abnormalities of the flagella (MMAF), a subtype of asthenoteratozoospermia, characterized by a mosaic of abnormalities of the flagellar morphology, including absent, short, coiled, bent and/or irregular flagella, is almost always caused by genetic defects (*Touré et al., 2021*; *Wang et al., 2020*). To date, more than 40 genes have been identified as pathogenic genes of MMAF, but these genes can only explain approximately 60% of MMAF-affected cases (*Wang et al., 2022*; *Lu et al., 2021*; *Houston et al., 2021*; *Jiao et al., 2021*). Therefore, the genetic basis of the remaining cases is still unknown.

The motility of a sperm is driven by its rhythmically beating flagella, and at the center of the flagella lies a conserved axonemal structure, containing the '9+2' microtubular arrangement: a ring of nine microtubule doublets (MTDs) surrounding a central pair (CP) of singlet microtubules. Each MTD consists of an A tubule and a B tubule, and the outer (ODA) and inner (IDA) dynein arms are anchored along the A tubule (*Leung et al., 2023*). The ODA and IDA are ATPase-based protein complexes that drive the movement between the A tubule and the neighboring B tubule of the next doublet, producing the original force for sperm motility (*Burgess et al., 2003*; *Linck et al., 2016*). Structural and functional abnormalities of the ODA and IDA have been demonstrated to cause male infertility associated with asthenozoospermia and/or asthenoteratozoospermia (*Touré et al., 2021*; *Gunes et al., 2020*; *Sironen et al., 2020*).

The dynein axonemal heavy chain (DNAH) family comprises a series of proteins (DNAH1–3, DNAH5–12, and DNAH17) that are precisely assembled with other axonemal dynein motor proteins in the ODAs and IDAs of sperm flagella and motile cilia (*King, 2016*; *Walton et al., 2023*; *Aprea et al., 2021*). In humans, DNAH1, DNAH2, DNAH6, DNAH7, DNAH8, DNAH10, DNAH12, and DNAH17 are highly expressed in the testis, and deficiency of these proteins has been demonstrated to cause MMAF-associated asthenoteratozoospermia (*Ben Khelifa et al., 2014*; *Hwang et al., 2021*; *Tu et al., 2019*; *Gao et al., 2022*; *Liu et al., 2020*; *Tu et al., 2021*; *Li et al., 2021*; *Whitfield et al., 2019*). DNAH3 is an evolutionarily conserved IDA-associated protein and is highly expressed in testes of humans and mice (*Chapelin et al., 1997*). Deficient DNAH3 has been shown to impair sperm motility in *Drosophila* and cattle (*Karak et al., 2015*; *Modiba et al., 2022*). In humans, *DNAH3* has been identified as a novel breast cancer candidate gene (*Hamdi et al., 2018*). However, the role of DNAH3 in male reproduction in humans and mice remains largely unknown.

In the present study, we identified four biallelic variations in *DNAH3* in four unrelated Han Chinese patients with asthenoteratozoospermia using whole-exome sequencing (WES). The spermatozoa of the patients showed extremely reduced sperm motility and a high proportion of sperm tail defects characterized by the MMAF phenotype. We further generated *Dnah3* knockout (KO) mice, and the male KO mice expectedly showed aberrations in sperm movement, flagellar IDA, and mitochondrion. Moreover, the absence of DNAH3 led to decreased expression of other IDA-associated proteins, including DNAH1, DNAH6 and DNALI1. Importantly, good outcomes of intracytoplasmic sperm injection (ICSI) treatment were observed in *DNAH3*-deficient patients and *Dnah3* KO mice. This study

revealed *DNAH3* as a novel pathogenic gene of asthenoteratozoospermia, and the findings provide valuable suggestions for the clinical diagnosis and treatment of male infertility.

## Results

### Identification of biallelic pathogenic variants of *DNAH3* in four unrelated infertile men

In the present study, we employed whole-exome sequencing (WES) to identify potential candidate variants associated with primary asthenoteratozoospermia. After comprehensive filtering and screening, we identified 98, 101, 67, and 91 candidate variants for Patient 1, Patient 2, Patient 3, and Patient 4, respectively (*Figure 1—source data 1*). To refine these candidate variants, we excluded those whose corresponding genes were not expressed in the human or mouse testis, were associated with diseases unrelated to male infertility, or were monoallelic variants. Ultimately, only bi-allelic variants in *DNAH3* (NG_052617.1, NM_017539.2, NP_060009.1) remained, suggesting as the pathogenic variants responsible for the infertility of the patients: a compound heterozygous mutation of c.3590C>T (p.Pro1197Leu) and c.3590C>G (p.Pro1197Arg) in Patient 1, a homozygous missense mutation of c.4837G>T (p.Ala1613Ser) in Patient 2, a compound heterozygous mutation of c.5587del (p.Leu1863*) and c.10355C>T (p.Ser3452Leu) in Patient 3, and a compound heterozygous mutation of c.2314C>T (p.Arg772Trp) and c.4045G>A (p.Asp1349Asn) in Patient 4 (*Figure 1A*). Importantly, routine semen analysis revealed that all patients showed extremely reduced sperm motility and a high proportion of sperm tail defects (*Table 1*). These variants either were not recorded or had an extremely low frequency in East Asian population in multiple public population databases, including the ExAC browser, GnomAD and the 1000 Genomes Project, and were predicted to be potentially deleterious by SIFT (https://sift.bii.a-star.edu.sg/), PolyPhen-2 (http://genetics.bwh.harvard.edu/pph2/), MutationTaster (https://www.mutationtaster.org/), and CADD (https://cadd.gs.washington.edu/; *Table 2*; *Ng and Henikoff, 2003*; *Adzhubei et al., 2010*; *Schwarz et al., 2014*; *Rentzsch et al., 2019*). Next, Sanger sequencing confirmed these variants in the probands, and their fertile parents carried the heterozygous variants (*Figure 1A*). Moreover, the variant sites are localized in several domains of the DNAH3 protein and are highly conserved across species (*Figure 1B*).

Strikingly, immunofluorescence staining revealed that DNAH3 was exclusively resided in the tail and concentrated in the midpiece of control sperm. However, the fluorescence signal of DNAH3 was hardly detected in the patients' spermatozoa (*Figure 1C*). Additionally, subsequent western blotting analysis yielded consistent results with immunofluorescence staining, indicating that these variants led to disrupted expression of DNAH3 (*Figure 1D*). These results suggested that biallelic variants in *DNAH3* disrupted DNAH3 expression and might be responsible for the infertility of the four patients.

### Asthenoteratozoospermia phenotype is observed in patients with *DNAH3* variants

We next investigated the aberrant sperm morphology of the patients using Papanicolaou staining and SEM analysis. Notably, the tails of sperm from the patients exhibited a typical phenotype associated with MMAF, including coiled, short, bent, irregular, and/or absent flagella (*Figure 2A and B*, *Figure 2—figure supplement 1*). In addition, a fraction of defects in the sperm head were also present in the patients' sperm (*Figure 2B*).

TEM was employed to determine the ultrastructure of the sperm from the patients. Compared to the integrated and well-organized '9 + 2' axonemal arrangement of the sperm flagella from the normal control, spermatozoa from the patients showed absent or disordered CPs, MTDs, and outer dense fibers (ODFs) in different regions of the flagella (*Figure 3A*, *Figure 3—figure supplement 1*). Interestingly, the IDAs of sperm flagella of the patients were hardly captured compared to the control (*Figure 3A*). Additionally, in the midpiece of sperm flagella of the patients, dissolved mitochondrial material was also observed evidently under TEM (*Figure 3A*). We next conducted immunofluorescence staining to label the mitochondria of patients' sperm with TOM20, a subunit of the mitochondrial import receptor. Remarkably, in contrast to the robust TOM20 signals observed in the normal control, the TOM20 signals in the sperm from the patients were considerably diminished, indicating a disrupted mitochondrial function (*Figure 3B*). Together, these data suggested that *DNAH3* may

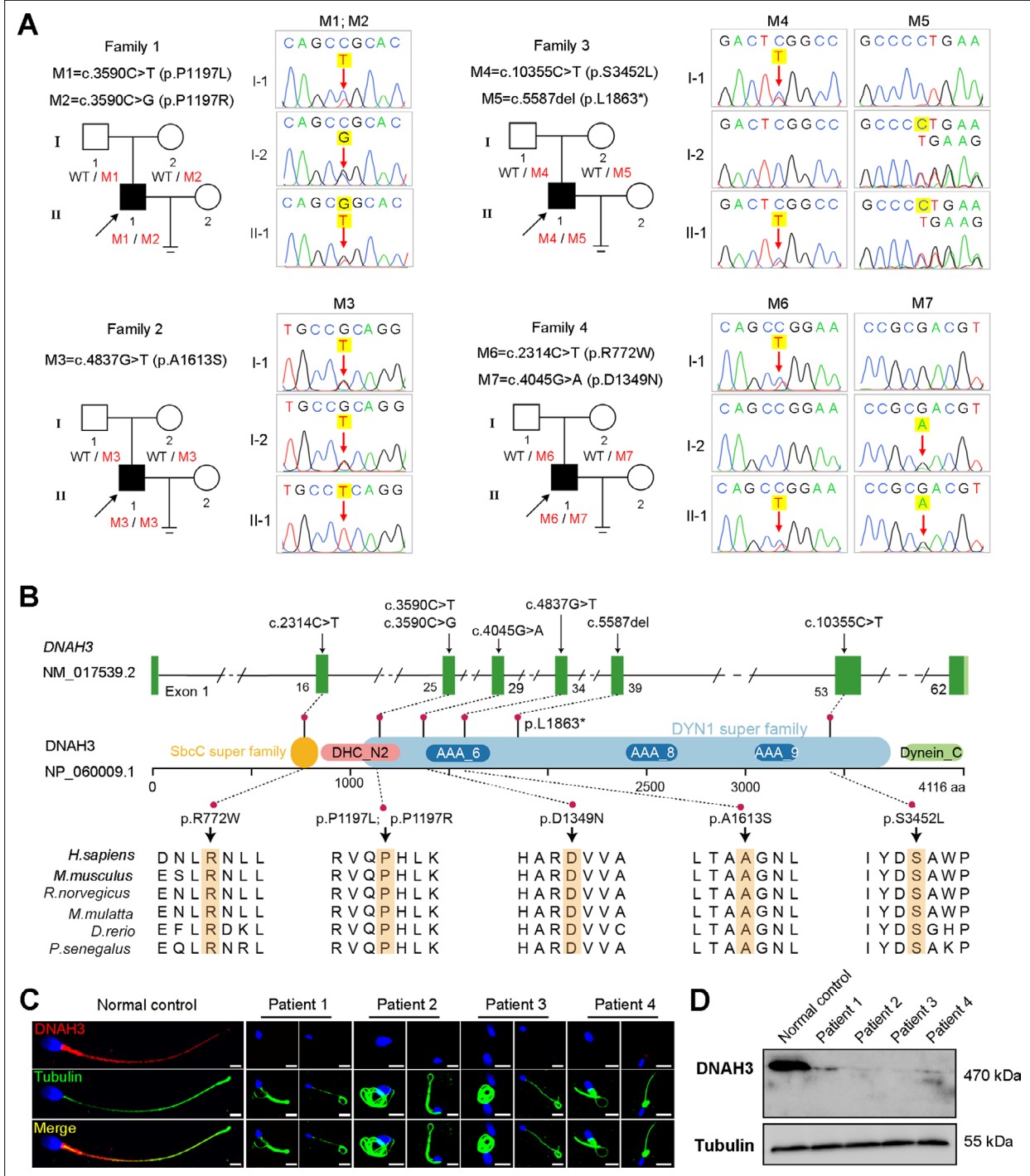

**Figure 1.** Identification of biallelic pathogenic variants in *DNAH3* from four unrelated infertile families. (**A**) Pedigrees of four families affected by *DNAH3* variants (M1–M7). Black arrows indicate the probands in these families. (**B**) Location of the variants and conservation of affected amino acids in DNAH3. Black arrows indicate the position of the variants. (**C**) Immunofluorescence staining of DNAH3 in sperm from the patients and normal control. Red, DNAH3; green, α-Tubulin; blue, DAPI; scale bars, 5 μm. (**D**) Western blotting analysis of DNAH3 expressed in spermatozoa from the patients and normal control.

The online version of this article includes the following source data for figure 1:

**Source data 1.** Summary of whole exome sequencing and the candidate variants identified.

**Source data 2.** Primers for Sanger sequencing.

**Source data 3.** PDF file containing original western blotting for *Figure 1D*.

**Source data 4.** Original files for western blotting analysis displayed in *Figure 1D*.

**Table 1.** Semen analysis of the patients in the present study.

| | | Patient 1 | Patient 2 | Patient 3 | Patient 4 | Reference |
|---|---|---|---|---|---|---|
| Semen parameters | Semen volume (ml) | 4.3 | 3.3 | 0.8 | 4.6 | ≥1.5 |
| | Semen concentration ($10^6$ /ml) | 2.0 | 0.5 | 11.0 | 21.0 | ≥15.0 |
| | Motility (%) | 6.0 | 3.0 | 0 | 0 | ≥40.0 |
| | Progressive motility (%) | 0 | 2.3 | 0 | 0 | ≥32.0 |
| | Normal (%) | 1.3 | 1.1 | 1.8 | 1.2 | ≥4.0 |
| Sperm morphology | Tail defects (%) | 91.3 | 87.5 | 96.5 | 97.5 | - |

-,not applicable.

function in sperm flagellar development, and loss-of-function variants were associated with MMAF in humans.

## DNAH3 is exclusively expressed in the sperm flagella of humans and mice

To further understand the function of DNAH3 in male reproduction, we explored the expression pattern of DNAH3 in humans and mice. qPCR results revealed that *Dnah3* was predominantly expressed in the mouse testis (*Figure 4—figure supplement 1A*). Moreover, when observing the expression of *Dnah3* in testes from mice at different postnatal days, we found that *Dnah3* expression was significantly elevated beginning on postnatal Day 22, peaked at postnatal Day 30, and maintained a stable expression level thereafter (*Figure 4—figure supplement 1B*). In addition, germ cells at different stages were isolated from the testes of humans and mice and were stained with anti-DNAH3 antibody. The results showed that DNAH3 was expressed in the cytoplasm of spermatocytes and spermatogonia and then obviously in the flagellum of early and late spermatids (*Figure 4—figure supplement 2A, B*). These expression data suggest that DNAH3 may play an important role in sperm flagellar development during spermatogenesis in humans and mice.

**Table 2.** Variants analysis of the patients in the present study.

| | | Patient 1 | | Patient 2 | Patient 3 | Patient 4 | | |
|---|---|---|---|---|---|---|---|---|
| Variant | cDNA mutation * | c.3590C>T | c.3590C>G | c.4837G>T | c.5587del | c.10355C>T | c.2314C>T | c.4045G>A |
| | Protein alteration | p.Pro1197Leu | p.Pro1197Arg | p.Ala1613Ser | p.Leu1863* | p.Ser3452Leu | p.Arg772Trp | p.Asp1349Asn |
| | Mutation type | Missense | Missense | Missense | Nonsense | Missense | Missense | Missense |
| Allele frequency | ExAC_EAS | 0.0001 | 0 | 0.004165 | 0 | 0.0006 | 0.0019 | 0.0065 |
| | GnomAD_EAS | 0.00005016 | 0 | 0.00277415 | 0 | 0.0008 | 0.002 | 0.007 |
| | 1000 Genomes Project_EAS | 0 | 0 | 0.0050 | 0 | 0 | 0.0040 | 0.0069 |
| Function prediction | SIFT | Deleterious | Deleterious | Deleterious | / | Tolerated | Deleterious | Deleterious |
| | Polyphen-2 | Probably damaging | Probably damaging | Probably damaging | / | Probably damaging | Probably damaging | Probably damaging |
| | Mutation Taster | Disease causing | Disease causing | Disease causing | / | Disease causing | Disease causing | Disease causing |
| | CADD † | 33 | 29.5 | 27.5 | / | 25.4 | 27.9 | 34 |

/,not applicable.
*NM_017539.2.
†score >4.0 is predicted to be damaging.

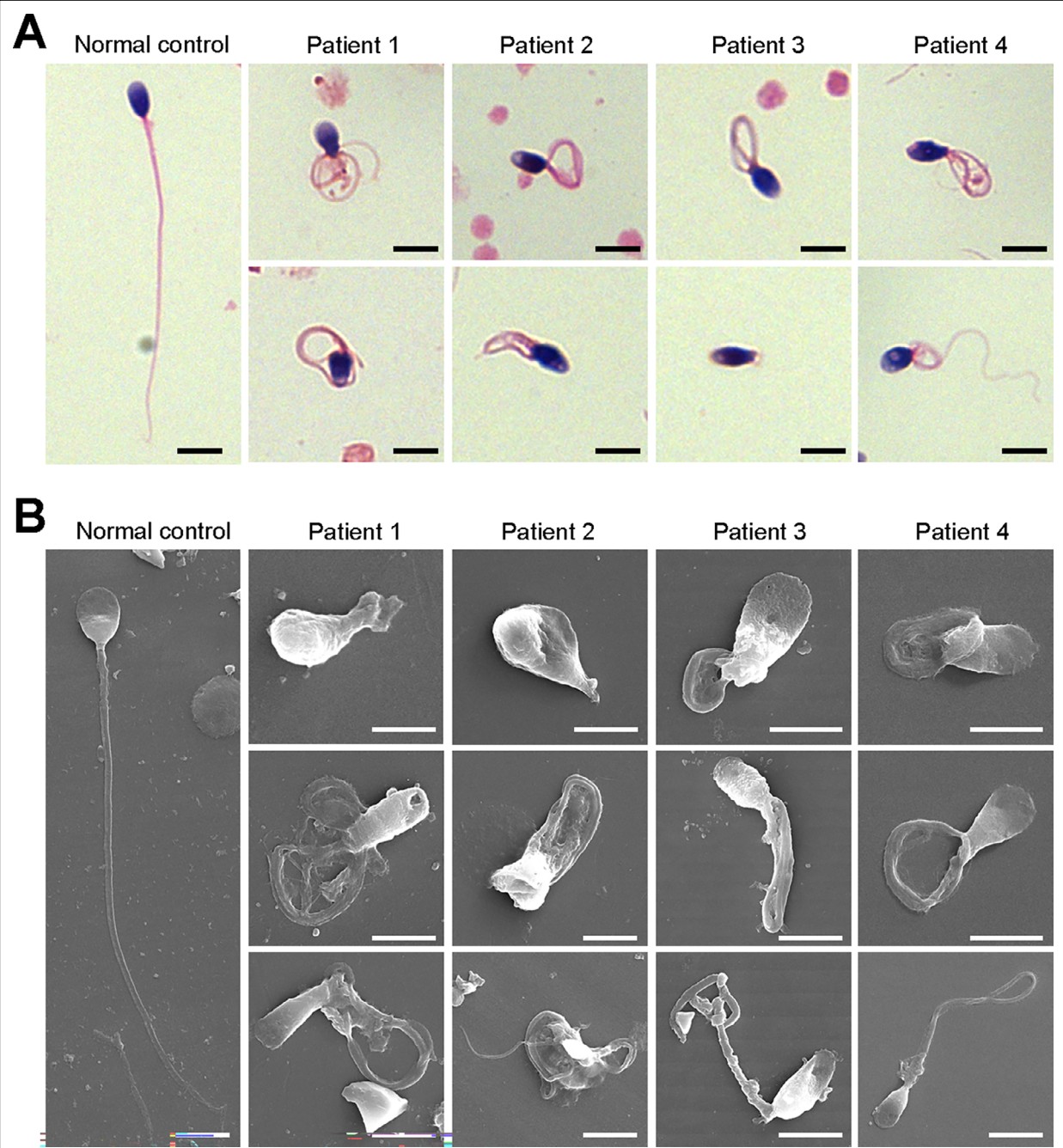

**Figure 2.** Defects in sperm morphology of the patients harboring *DNAH3* variants. (**A, B**) Abnormal sperm morphology was observed through Papanicolaou staining (**A**), and SEM analysis (**B**) compared to normal control. Scale bars, 5 μm.

The online version of this article includes the following source data and figure supplement(s) for figure 2:

**Figure supplement 1.** The histogram of various flagellar morphology in the normal control and patients.

**Figure supplement 1—source data 1.** The percentage distribution of various flagellar morphology in the normal control and patients.

## Deletion of *Dnah3* causes male infertility in mice

Considering the absent expression of DNAH3 in the patient sperm, we generated *Dnah3* KO mice using CRISPR–Cas9 technology to further confirm the essential role of DNAH3 in spermatogenesis (*Figure 4—figure supplement 3A*). PCR, qPCR, and immunofluorescence staining were used to confirm that *Dnah3* was null in KO mice (*Figure 4—figure supplement 3B–E*). The *Dnah3* KO

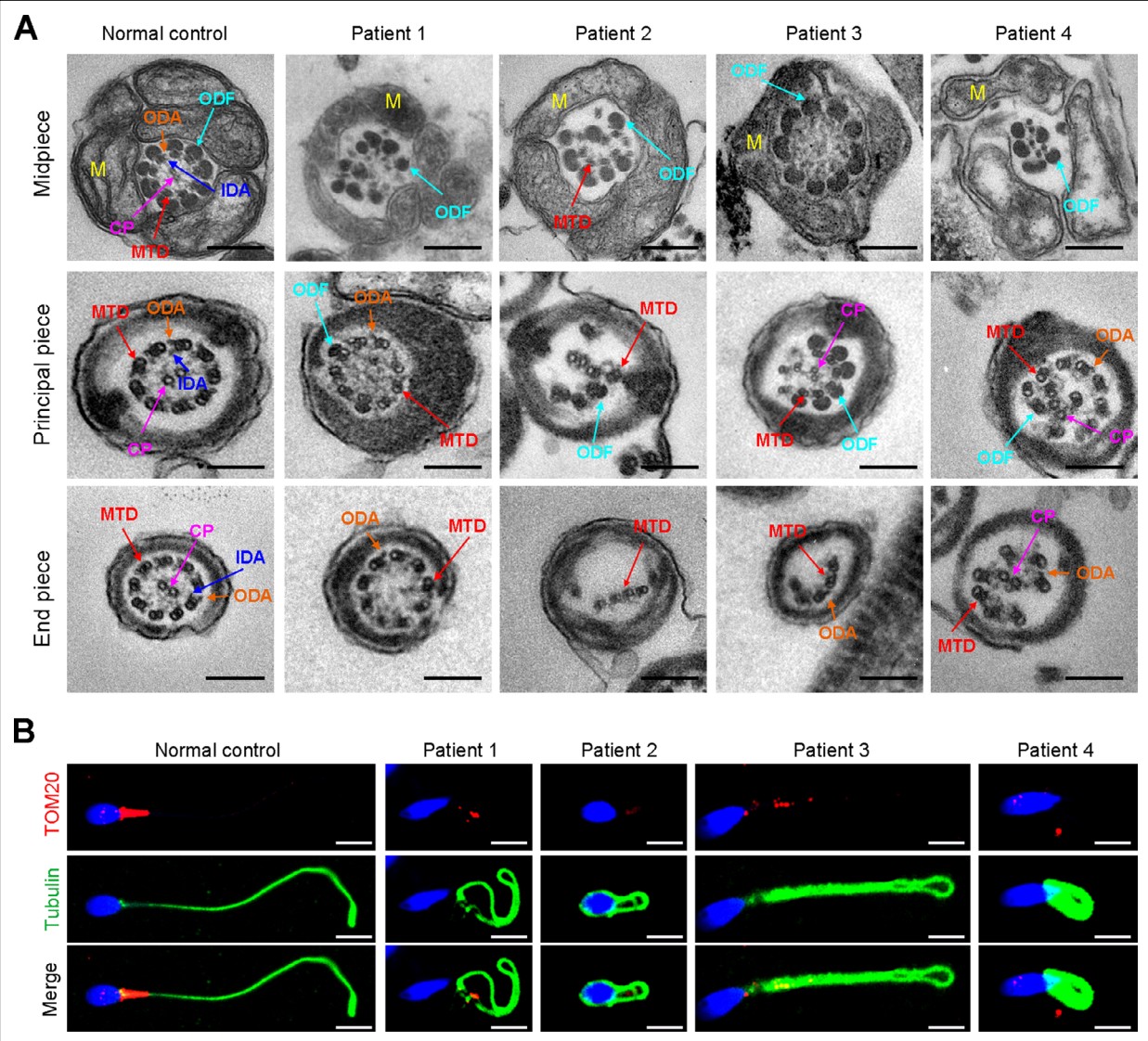

**Figure 3.** Ultrastructural and mitochondrial defects in sperm from infertile men with *DNAH3* variants. (**A**) TEM analysis of sperm obtained from a normal control and patients harboring *DNAH3* variants. Cross-sections of the midpiece, principal piece and endpiece of sperm from normal control showed the typical "9+2" microtubule structure, and an IDA and an ODA were displayed on the A-tube of each microtubule doublet. Cross-sections of the midpiece, principal piece and endpiece of sperm from the patients displayed absent or disordered CPs, MTDs and ODFs, as well as an evident missing of the IDAs in different pieces of the flagella. M, mitochondria sheath; ODF, outer dense fiber; MTD, microtubule doublets; CP, central pair; IDA, inner dynein arms; ODA, outer dynein arms. Scale bars, 200 nm. (**B**) Immunofluorescence staining of TOM20 in sperm from the patients and normal control. Red, TOM20; green, α-Tubulin; blue, DAPI; scale bars, 5 μm.

The online version of this article includes the following figure supplement(s) for figure 3:

**Figure supplement 1.** The percentage of aberrant ultrastructure in different cross-sections of sperm from the normal control and patients.

mice survived without any evident abnormalities in development and behavior. H&E staining further revealed that there were no histological differences in the lung, brain, eye, or oviduct between wild-type (WT) and *Dnah3* KO mice (*Figure 4—figure supplement 4A*). In addition, no obvious abnormalities in ciliary development were observed in these organs in KO mice compared to WT mice (*Figure 4—figure supplement 4B*). The *Dnah3* KO female mice were fertile with normal oocyte development (*Figure 4—figure supplement 5A*). However, the *Dnah3* KO male mice were completely infertile (*Figure 4A*). We next examined the testis and epididymis of *Dnah3* KO male mice to elucidate the etiology of infertility. There was no detectable difference in the testis/body weight ratio of *Dnah3* KO mice when compared to WT mice (*Figure 4—figure supplement 5B*).

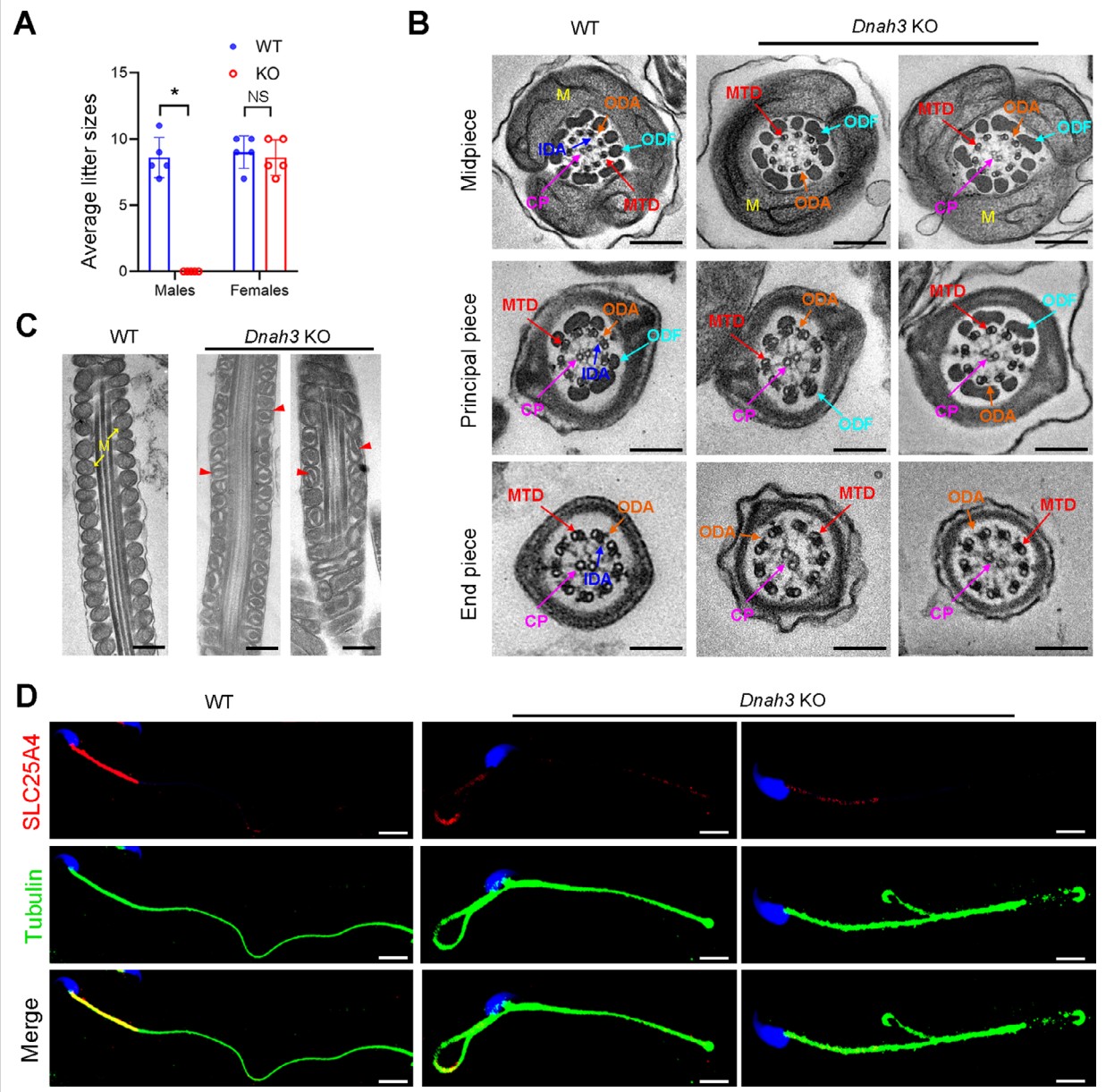

**Figure 4.** *Dnah3* KO male mice are infertile. (**A**) Fertility of *Dnah3* KO mice. The KO male mice were infertile (n=five biologically independent WT mice or KO mice; Student's *t* test; *, p<0.05; NS, no significance; error bars, s.e.m.). (**B**) TEM analysis of the cross-sections of spermatozoa from *Dnah3* KO mice revealed an obvious absence of IDAs in different pieces of the flagella compared to WT mice. M, mitochondrion sheath; ODF, outer dense fiber; MTD, microtubule doublet; CP, central pair; IDA, inner dynein arm; ODA, outer dynein arm. Scale bars, 200 nm. (**C**) Disrupted mitochondria were observed in spermatozoa tail from *Dnah3* KO mice by TEM analysis. The yellow arrows indicate the normal mitochondria. The red arrowheads indicate the dilated intermembrane spaces and dissolved mitochondrial material. M, mitochondrion sheath. Scale bars, 200 nm. (**D**) Immunofluorescence staining of SLC25A4 indicated impaired mitochondrial formation in *Dnah3* KO mice compared to WT mice. Red, SLC25A4; green, α-Tubulin; blue, DAPI; scale bars, 5 μm.

The online version of this article includes the following video, source data, and figure supplement(s) for figure 4:

**Figure supplement 1.** The expression of DNAH3 in mouse testis.

**Figure supplement 1—source data 1.** PDF file containing original gels for *Figure 4—figure supplement 1A, B*.

**Figure supplement 1—source data 2.** Original files for gel analysis displayed in *Figure 4—figure supplement 1A, B*.

**Figure supplement 2.** DNAH3 is expressed during spermatogenesis in mice and humans.

**Figure supplement 3.** Generation of *Dnah3* KO mice.

**Figure supplement 3—source data 1.** Primers for Sanger sequencing and qPCR.

*Figure 4 continued on next page*

*Figure 4 continued*

**Figure supplement 4.** Ciliary development of *Dnah3* KO mice.

**Figure supplement 5.** Fertility of *Dnah3* KO mice.

**Figure supplement 6.** Morphology and ultrastructure of sperm isolated from *Dnah3* KO mice.

**Figure 4—video 1.** CASA of sperm from WT mice.

https://elifesciences.org/articles/96755/figures#fig4video1

**Figure 4—video 2.** CASA of sperm from *Dnah3* KO mice.

https://elifesciences.org/articles/96755/figures#fig4video2

Moreover, subsequent computer-assisted sperm analysis (CASA) also showed that sperm isolated from the cauda epididymis were slightly decreased, and nearly all sperm were completely immobile (*Table 3*, *Figure 4—video 1* and *Figure 4—video 2*). Papanicolaou staining and SEM analysis revealed morphological defects in partial spermatozoa from *Dnah3* KO mice, including coiled, bent, and irregular flagella, as well as aberrant heads and acephalic spermatozoa (*Figure 4—figure supplement 6A, B*).

TEM was further utilized to evaluate the sperm flagellar ultrastructure of *Dnah3* KO mice. There were no obvious abnormalities of '9+2' microtube arrangement in most sperm from the *Dnah3* KO mice when compared to WT mice (*Figure 4B*, *Figure 4—figure supplement 6C*). However, in contrast to the clear display of an IDA and an ODA on the A-tube of each microtubule doublet in the sperm flagella of WT mice, the sperm flagella of *Dnah3* KO mice exhibited an absence of almost all the IDAs (*Figure 4B*, *Figure 4—figure supplement 6D*). In addition, the disrupted mitochondria of spermatozoa from *Dnah3* KO mice were also observed under TEM, as manifested by the dilated intermembrane spaces and dissolved mitochondrial material (*Figure 4B and C*, *Figure 4—figure supplement 6E*). We next performed immunofluorescence staining to label SLC25A4, which is responsible for the exchange of ATP and ADP across the mitochondrial inner membrane. Strikingly, compared to the

**Table 3.** Semen analysis using CASA in the *Dnah3* KO mice.

|  | WT | KO | p* value |
|---|---|---|---|
| **Semen parameters** | | | |
| Sperm concentration ($10^6$/ml) [†] | 112.32±18.26 | 105.17±11.15 | 0.059 |
| Motility (%) * | 71.56±3.97 | 4.37±1.15 | <0.01 |
| Progressive motility (%) * | 60.36±4.32 | 4.37±1.15 | <0.01 |
| **Sperm locomotion parameters** | | | |
| Curvilinear velocity (VCL) (µm/s)* | 67.54±6.79 | 9.07±1.22 | <0.01 |
| Straight-line velocity (VSL) (µm/s)* | 28.91±4.86 | 2.68±0.52 | <0.01 |
| Average path velocity (VAP) (µm/s)* | 39.02±5.31 | 3.85±0.82 | <0.01 |
| Amplitude of lateral head displacement (ALH) (µm)* | 0.71±0.03 | 0.13±0.04 | <0.01 |
| Linearity (LIN)* | 0.43±0.07 | 0.30±0.02 | 0.037 |
| Wobble (WOB,=VAP/VCL)* | 0.58±0.06 | 0.42±0.05 | 0.024 |
| Straightness (STR,=VSL/VAP) | 0.74±0.22 | 0.70±0.13 | 0.80 |
| Beat-cross frequency (BCF) (Hz)* | 4.86±0.12 | 0.73±0.08 | <0.01 |

*A significant difference, two-sided student's t-test. n=3 biologically independent WT mice or KO mice.

[†]Epididymides and vas deferens.

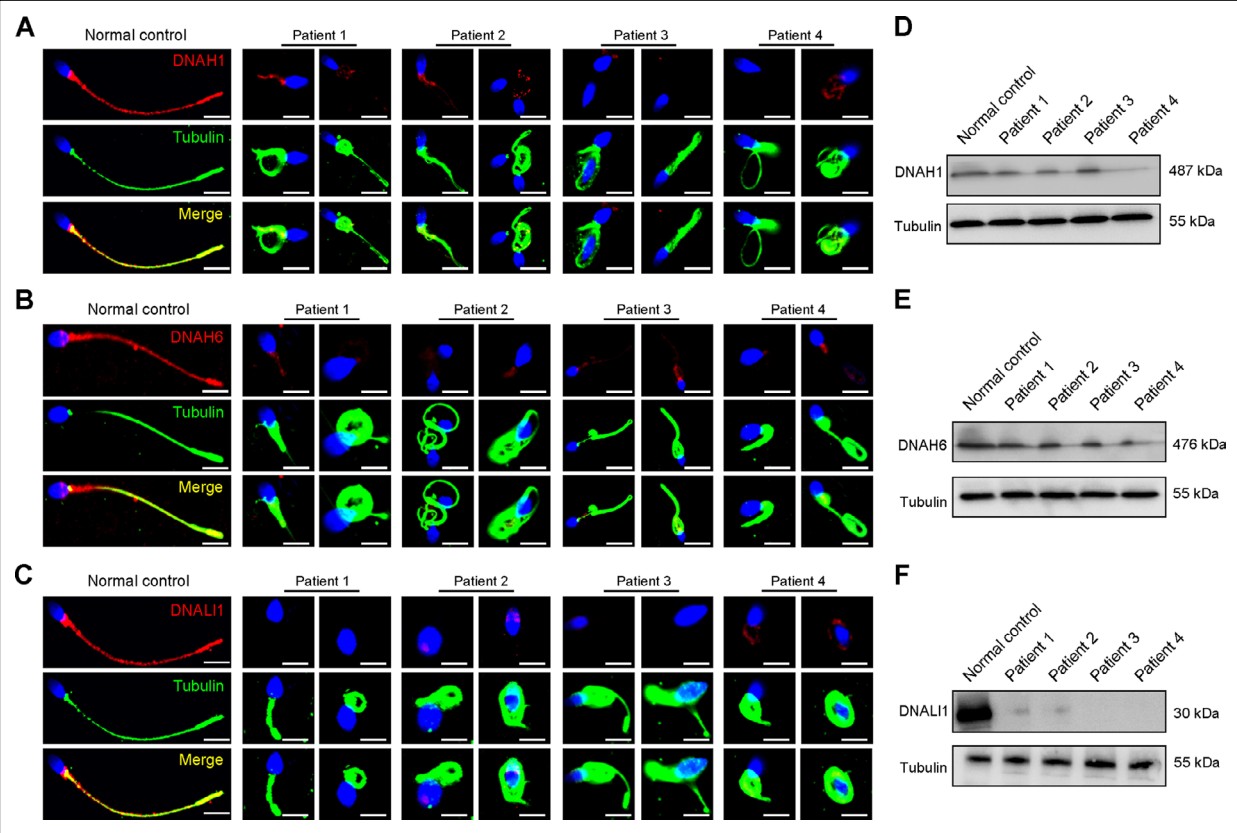

**Figure 5.** Immunofluorescence staining and western blotting analysis of IDA-associated proteins in spermatozoa obtained from normal control and patients with *DNAH3* variants. (**A – C**) Immunofluorescence staining of DNAH1 (**A**), DNAH6 (**B**) and DNALI1 (**C**) in spermatozoa from patients and normal controls. Red, DNAH1 in (**A**), DNAH6 in (**B**), DNALI1 in (**C**); green, α-Tubulin; blue, DAPI; scale bars, 5 μm. (**D – F**) Western blotting analysis of DNAH1(D), DNAH6 (**E**), DNALI1 (**F**) in sperm lysates from the patients and normal control.

The online version of this article includes the following source data and figure supplement(s) for figure 5:

**Source data 1.** PDF file containing original western blotting for *Figure 5D–F* and *Figure 6D–F*.

**Source data 2.** Original files for western blotting analysis displayed in *Figure 5D–F* and *Figure 6D–F*.

**Figure supplement 1.** Immunofluorescence staining of ODA-associated proteins in spermatozoa obtained from variants within *DNAH3* patients.

---

bright fluorescence signals in the midpiece of WT sperm, the signals *Dnah3* KO were significantly diminished (*Figure 4D*), indicating impaired mitochondrial function. Collectively, DNAH3 is essential for spermatogenesis, and its deficiency seriously damages the sperm motility and IDAs in both humans and mice.

## DNAH3 deficiency impairs IDAs related to the reduction of IDA-associated proteins

Considering the disrupted IDAs revealed by TEM analysis in both our patients and *Dnah3* KO mice, we speculated whether the defective IDAs were attributed to the decreased expression of the key IDA-associated proteins. The immunofluorescence data showed that DNAH1/DNAH6 and DNALI1, corresponding to the heavy and light intermediate chains of the IDAs (*Walton et al., 2023*), respectively, were almost invisible along the sperm flagella of the patients when compared to control (*Figure 5A–C*). Consistent results were obtained in our subsequent western blotting analysis of sperm lysates from the patients (*Figure 5D–F*), indicating that DNAH3 may manipulate the assembly of IDA through regulating the expression of IDA-associated proteins. In contrast, DNAH8/DNAH17 and DNAI1, corresponding to the heavy and intermediate chains of ODAs (*Whitfield et al., 2019*), were readily detectable in the patients' sperm flagella and were comparable to the control (*Figure 5—figure supplement 1A–C*), suggesting that DNAH3 may not regulate the expression of ODA-associated proteins. We also performed immunofluorescence staining and western blotting analysis of DNAH1,

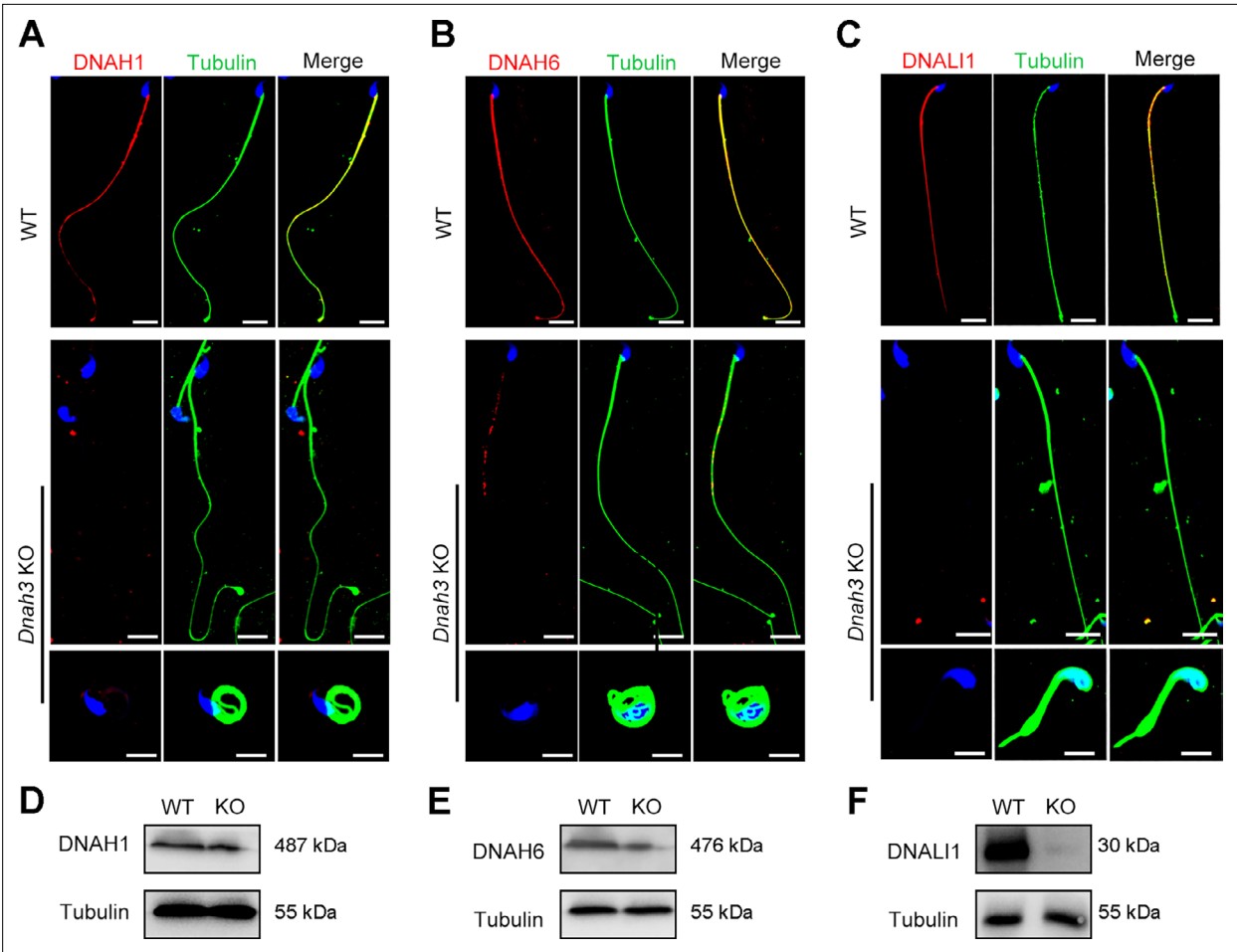

**Figure 6.** Immunofluorescence staining and western blotting analysis of IDA-associated proteins in spermatozoa from WT and *Dnah3* KO mice. (**A – C**) Immunofluorescence staining of DNAH1 (**A**), DNAH6 (**B**) and DNALI1 (**C**) in spermatozoa from *Dnah3* KO and WT mice. Red, DNAH1 in (**A**), DNAH6 in (**B**), DNALI1 in (**C**); green, α-Tubulin; blue, DAPI; scale bars, 5 μm. (**D – F**) Western blotting analysis of DNAH1(D), DNAH6 (**E**) and DNALI1 (**F**) in spermatozoa lysates from *Dnah3* KO and WT mice.

The online version of this article includes the following figure supplement(s) for figure 6:

**Figure supplement 1.** Immunofluorescence staining of ODA-associated proteins in spermatozoa of *Dnah3* KO and WT mice.

DNAH6, DNALI1, DNAH8, DNAH17 and DNAI1 on sperm from *Dnah3* KO mice, and the results observed were consistent with those of the patients (*Figure 6A–F*, *Figure 6—figure supplement 1A–C*). These findings suggested that other IDA-associated proteins might be downstream effectors of DNAH3, which needs more future research.

## ICSI treatment of humans with *DNAH3* variants and *Dnah3* KO mice

ICSI treatment has been reported to be effective in asthenoteratozoospermia-associated infertility (*Tsai et al., 2011*; *Colpi et al., 2018*). ICSI cycles were attempted for the patients after written informed consent was obtained. The female partners all had normal basal hormone levels and underwent a long gonadotrophin-releasing hormone agonist protocol (*Table 4*). The wife of Patient 1 underwent one ICSI attempt. A total of 21 metaphase II (MII) oocytes were retrieved and microinjected, of which 17 oocytes were successfully fertilized (17/21, 80.95%) and cleaved (17/17, 100%). Thirteen Day 3 (D3) embryos were formed, six of which developed into blastocysts (8/13, 61.54%) after standard embryo culture. Two blastocysts were transferred, one of which was implanted. She eventually achieved clinical pregnancy, and the pregnancy is ongoing (*Table 4*). The partner of Patient 2 underwent two ICSI attempts. In her first ICSI attempt, six MII oocytes were retrieved, of which three were fertilized (3/6, 50%) and cleaved (3/3, 100%). After standard embryo culture, two D3 embryos were formed

**Table 4.** Outcomes of ICSI treatment in the patients with *DNAH3* mutations.

| | Patient | Patient 1 | Patient 2 | | Patient 3 | Patient 4 | | | |
|---|---|---|---|---|---|---|---|---|---|
| | Female age (y) | 24 | 30 | | 30 | 36 | | | |
| Subjects | Length of primary infertility history (y) | 6 | 1 | | 3 | 8 | | | |
| | FSH (IU/L) | 7.1 | 8.5 | | 3.49 | 4.3 | | | |
| | LH (IU/L) | 4.44 | 2.5 | | 4.2 | 2.6 | | | |
| | E2 (pg/mL) | 83 | 50 | | 43.49 | 68 | | | |
| Basal hormones | Prog (ng/mL) | 0.2 | 0.6 | | 0.3 | 0.3 | | | |
| | | Cycle 1 | Cycle 1 | Cycle 2 | Cycle 1 | Cycle 1 | Cycle 2 | Cycle 3 | Cycle 4 |
| | E2 level on the trigger day (pg/ml) | 3366 | 1519.6 | 1582.4 | >5,000 | 4440 | 5000 | 3152 | 2206 |
| | No. of follicles ≥14 mm on the trigger day | 15 | 6 | 5 | 20 | 13 | 12 | 14 | 11 |
| | No. of follicles ≥18 mm on the trigger day | 10 | 3 | 4 | 8 | 11 | 9 | 7 | 5 |
| ICSI Cycles | No. of oocytes retrieved | 24 | 6 | 5 | 25 | 16 | 20 | 21 | 14 |
| | No. of MII oocytes | 21 | 6 | 5 | 20 | 12 | 13 | 8 | 7 |
| | Fertilization rate (%) | 80.95 (17/21) | 50 (3/6) | 100 (5/5) | 95 (19/20) | 41.67 (5/12) | 46.15 (6/13) | 50 (4/8) | 42.86 (3/7) |
| | Cleavage Rate (%) | 100 (17/17) | 100 (2/2) | 100 (5/5) | 100 (19/19) | 100 (5/5) | 100 (6/6) | 100 (4/4) | 66.7 (2/3) |
| | Available D3 embryos | 13 | 2 | 5 | 15 | 2 | 2 | 0 | 0 |
| ICSI progress | Blastocyst formation rate (%) | 61.5 (8/13) | 0 | 66.7 (2/3) | 66.7 (10/15) | - | - | - | - |
| | No. of embryos transferred | 2 blastocysts | 0 | 2 D3 | 1 blastocyst | 2 D3 | 2 D3 | - | - |
| Clinical outcomes | Implantation rate (%) | 50 (1/2) | 0 | 0 | 100 (1/1) | 0 | 0 | - | - |
| | Clinical pregnancy | Yes | No | No | Yes | - | - | - | - |
| | No. of live birth | ngoing | - | - | ngoing | - | - | - | - |

-,not applicable.

and transferred. However, this ICSI failed because no embryos were implanted. In her second ICSI attempt, all five MII oocytes were fertilized and cleaved (5/5, 100%). Five D3 embryos were obtained, of which two were transferred, but no embryos were implanted. The remaining three D3 embryos were cultured continuously, and two available blastocysts were formed and kept to be transferred in the future (*Table 4*). The partner of Patient 3 underwent one ICSI attempt. Of the 20 MII oocytes retrieved, 19 oocytes were fertilized (19/20, 95.0%) and cleaved (19/19, 100%). Fifteen D3 embryos were obtained, and 10 developed into available blastocysts (10/15, 66.7%). One blastocyst was transferred and implanted. She achieved clinical pregnancy, and the pregnancy is ongoing (*Table 4*). The wife of Patient 4 underwent four failed ICSI attempts. In her first two ICSI attempts, 13 and 12 MII oocytes were retrieved, of which five (5/12, 41.67%) and six (6/13, 46.15%), respectively, were fertilized and cleaved. Two available D3 embryos were obtained and transferred in both ICSI attempts, but no embryos were implanted. In her third ICSI attempt, of the eight MII oocytes retrieved, four (4/8, 50%) were fertilized and cleaved (4/4, 100%). However, no available D3 embryos were acquired. In her last ICSI attempt, seven MII oocytes were retrieved, of which three were fertilized (3/7, 42.68%) and two were cleaved (2/3, 66.7%), but no available D3 embryos were formed (*Table 4*). The vivid embryonic development of the partner of Patient 1 and Patient 3 after ICSI treatment was shown in *Figure 7A*.

We also carried out ICSI treatment on *Dnah3* KO male mice. Strikingly, favorable outcomes of ICSI were obtained in *Dnah3* KO male mice. After injection of spermatozoa from *Dnah3* KO male mice,

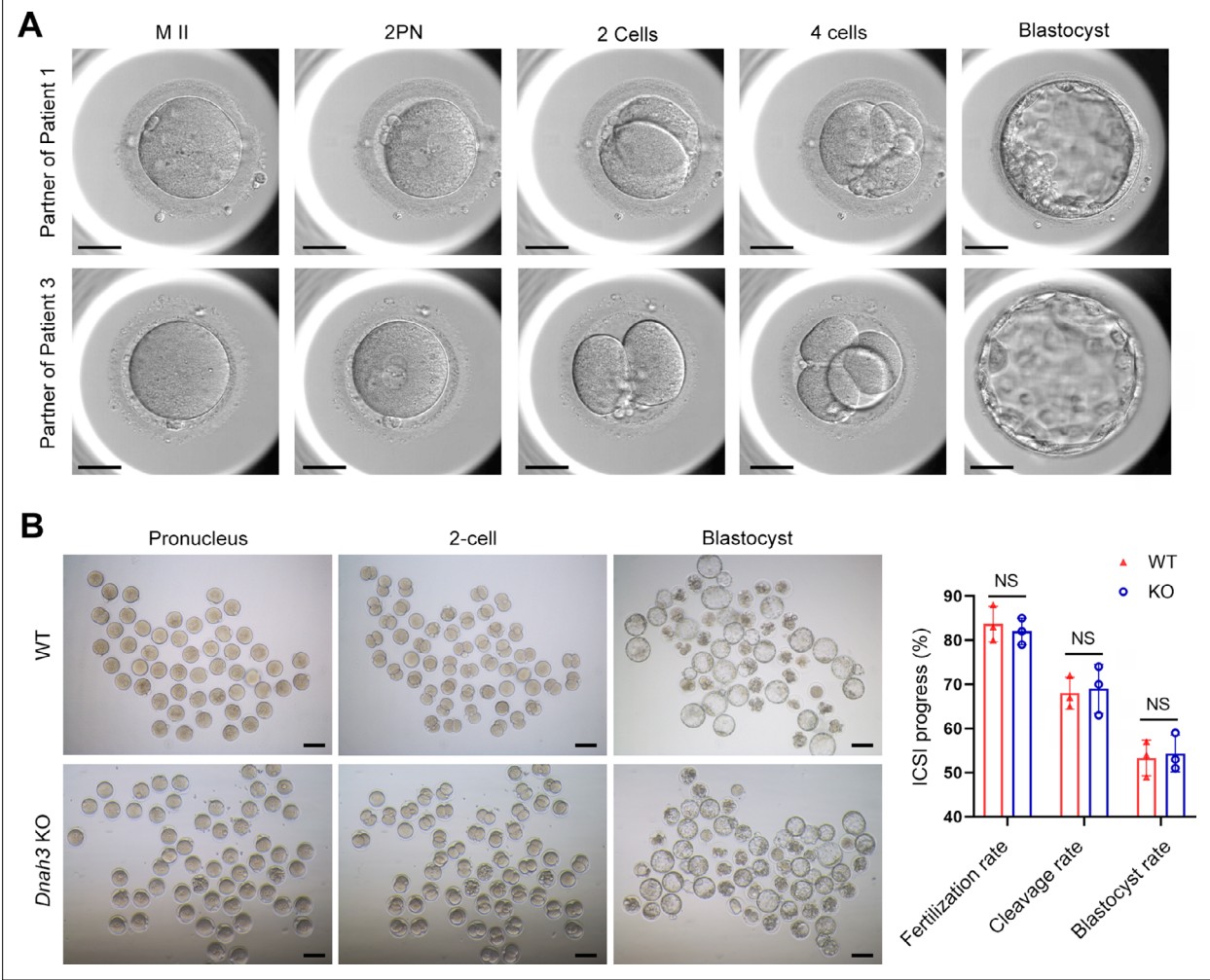

**Figure 7.** ICSI outcomes of *DNAH3*-deficient patients and *Dnah3* KO mice. (**A**) The embryonic development of Patient 1 and Patient 3 after ICSI treatment. MII, metaphase II; PN, pronucleus; scale bars, 40 μm. (**B**) There was no difference in the fertilization rate or 2 cell and blastocyst embryo formation rates between the *Dnah3* KO and WT groups (n=three biologically independent WT mice or KO mice; Student's *t* test; NS, no significance; error bars, s.e.m.).

pronuclei were observed in most embryos in both the KO and WT groups, indicating a normal fertilization rate (*Figure 7B*). There was no difference in the percentage of 2 cell and blastocyst-stage embryos between the KO and WT groups (*Figure 7B*). Collectively, we observed successful ICSI outcomes in two out of four DNAH3-deficient patients and *Dnah3* KO male mice and therefore suggested ICSI as an optional treatment for infertile men harboring biallelic pathogenic variants in *DNAH3*, and the additional female risk factors for infertility should not be excluded in the failed patients.

## Discussion

In the present study, we identified pathogenic variants in *DNAH3* in unrelated infertile men with asthenoteratozoospermia. These variations resulted in the almost absence of DNAH3 and sharply decreased the expression of other IDA-associated proteins, including DNAH1, DNAH6 and DNALI1. Combined with similar findings in *Dnah3* KO mice, we demonstrated that DNAH3 is fundamental for male fertility. Moreover, we suggest that ICSI might be a favorable treatment for male infertility caused by DNAH3 deficiency. Our findings identify a function for DNAH3 in male reproduction in humans and mice and may provide a new view on the clinical practice of male infertility.

Recently, Meng et al. reported *DNAH3* mutations in asthenoteratozoospermia affected patients, revealing multiple morphological defects in sperm tail (*Meng et al., 2024*). Moreover, ultrastructural

abnormalities of the flagellar axoneme in the patients were evident in these patients, characterized by a disrupted '9+2' arrangement and the notable absence of IDAs (*Meng et al., 2024*). Additionally, they generated *Dnah3* KO mice, which were infertile and exhibited moderate morphological abnormalities (*Meng et al., 2024*). While the '9+2' microtubule arrangement in the flagella of their *Dnah3* KO mice remained intact, the IDAs on the microtubules were partially absent (*Meng et al., 2024*). In our study, we observed similar phenotypic differences between *DNAH3*-deficient patients and *Dnah3* KO mice. Both studies suggest that *DNAH3* may play crucial yet distinct roles in human and mouse male reproduction.

However, there are notable differences between the two studies. Firstly, the phenotypes of *Dnah3* KO mice showed slight differences. Meng et al. generated two *Dnah3* KO mouse models (KO1 and KO2), and both of which exhibited significantly higher sperm motility and progressive motility than in our study (*Meng et al., 2024*), where nearly all sperm were completely immobile. Furthermore, their *Dnah3* KO2 mice displayed motility comparable to WT mice and retained partial fertility (*Meng et al., 2024*). We speculate that these differences may be attributed to variations in mouse genetic background or the presence of a truncated DNAH3 protein resulting from specific knockout strategies. Secondly, we conducted additional research and uncovered novel findings. We revealed that male infertility caused by *DNAH3* mutations follows an autosomal recessive inheritance pattern, as confirmed through Sanger sequencing of the patients' families. We also discovered the dynamic expression and localization of DNAH3 during spermatogenesis in humans and mice through immunofluorescent staining. Initially, DNAH3 was expressed in the cytoplasm of spermatogonia and spermatocytes, and then it clearly transferred into the flagellum of early and late spermatids. We further found that DNAH3 deficiency had no impact on ciliary development in the oviduct or on oogenesis in mice, resulting in normal female fertility. Moreover, in the absence of DNAH3 in both humans and mice, the expression of IDA-associated proteins, including DNAH1, DNAH6, and DNALI1, was decreased, while the expression of ODA-associated proteins remained unaffected, indicating that DNAH3 is involved in sperm axonemal development, specifically through its role in the assembly of IDAs. Collectively, our study corroborates the findings of Meng et al., and provides additional unique insights, comprehensively elucidating the critical role of DNAH3 in human and mouse spermatogenesis.

Primary ciliary dyskinesia (PCD, MIM: 244400) is a genetic disorder affecting at least one in 7554 individuals (*Hannah et al., 2022*). The most common symptoms of PCD are recurrent infections in airways due to malfunction of the motile cilia that are responsible for mucus clearance (*O'Connor et al., 2023*). It has been suggested that male infertility associated with sperm defects is highly prevalent (up to 75%) among individuals with PCD (*Vanaken et al., 2017*). Axonemal defects caused by variants within *DNAH* family members, including *DNAH5*, *DNAH6*, *DNAH7*, *DNAH9* and *DNAH11*, are causative factors for PCD (*Hornef et al., 2006*; *Guan et al., 2021*; *Peng et al., 2022*). Moreover, deficiency in these PCD-causing *DNAH*s has also been associated with male infertility (*Jiao et al., 2021*; *Sironen et al., 2020*; *Tu et al., 2019*; *Gao et al., 2022*; *Fliegauf et al., 2005*; *Fassad et al., 2018*; *Zuccarello et al., 2008*). Additionally, other *DNAH*s, such as *DNAH1*, *DNAH2*, *DNAH8*, *DNAH10*, *DNAH12*, and *DNAH17*, are suggested to be pathogenic genes of isolated male infertility (*Ben Khelifa et al., 2014*; *Hwang et al., 2021*; *Liu et al., 2020*; *Li et al., 2021*; *Whitfield et al., 2019*). These phenotype–genotype correlations may be attributed to the fact that ciliary and flagellar axonemes have cell-type-specific or cell-type-enriched DNAHs (*Wallmeier et al., 2020*). DNAH3 resides in the IDA and is expressed in testis and ciliary tissues, including the lung, brain, eye, and oviduct. However, despite its presence in these tissues, the relationship between deficient DNAH3 and disease is unclear to date. Intriguingly, in our study, none of the patients with *DNAH3* deficiency reported experiencing any of the principal symptoms associated with PCD. Additionally, our *Dnah3* KO mice exhibited normal ciliary development in the lung, brain, eye, and oviduct. Similarly, Meng et al. did not mention any PCD symptoms in their *DNAH3*-deficient patients, and their *Dnah3* KO mice also demonstrated normal ciliary morphology in the trachea and brain (*Meng et al., 2024*). These combined observations suggest that DNAH3 may play a more important role in sperm flagellar development than in other motile cilia functions. Given that DNAH3 is expressed in ciliary tissues, its role in these tissues remains intriguing and could be elucidated through sequencing of larger cohorts of individuals with PCD.

ICSI has been an efficient treatment for male infertility (*Pan et al., 2018*; *Esteves et al., 2018*). However, the outcomes of ICSI for male infertility caused by variants in different *DNAH* genes are

variable. It has been demonstrated that infertile males with variants in *DNAH1*, *DNAH2*, *DNAH7*, and *DNAH8* have a favorable prognosis (*Liu et al., 2020*; *Long et al., 2023*; *Wambergue et al., 2016*; *Li et al., 2019*; *Gao et al., 2021*; *Wei et al., 2021*), while patients with variants in *DNAH17* have poor outcomes after ICSI treatment (*Whitfield et al., 2019*; *Zheng et al., 2021*). Meanwhile, the ICSI outcomes in male infertility caused by *DNAH6* variants may depend on the specific mutation or be controversial (*Tu et al., 2019*; *Huang et al., 2023*; *Li et al., 2018*). The patients with *DNAH3* mutations in our study experienced different clinical outcomes of ICSI treatment. The partners of Patient 1 and Patient 3 achieved clinical pregnancy. The wives of Patient 2 and Patient 4 obtained favorable fertilization and cleavage rates but experienced no clinical pregnancy due to the nonimplantation of the transferred embryos. Remarkably, despite the diverse variants within *DNAH3* observed in the four patients, all variants led to a complete absence of DNAH3 expression. Additionally, we did not identify any pathogenic variants that associated with fertilization failure and early embryonic development in the two patients with failed ICSI outcomes. Therefore, these different ICSI outcomes might be attributed to additional unexplained factors from the female partners. Importantly, in the study from Meng et al., one patient carrying *DNAH3* variants received ICSI treatment, and the partner obtained clinical pregnancy (*Meng et al., 2024*). Combined with the successful ICSI outcomes observed in *Dnah3* KO mice, we suggest ICSI as an optimized treatment for infertile men carrying variants in *DNAH3*. More cases are needed to precisely estimate the prevalence of *DNAH3* mutations and determine a prognosis for ICSI treatments.

In conclusion, our study revealed an unexplored role of DNAH3 in male reproduction in humans and mice, suggesting *DNAH3* as a novel causative gene for human asthenoteratozoospermia. Moreover, ICSI is as an optimized treatment for infertile men with *DNAH3* variants. This study expands our knowledge of the relationship between DNAH proteins and disease, facilitating genetic counseling and clinical treatment of male infertility in the future.

## Materials and methods

**Key resources table**

| Reagent type (species) or resource | Designation | Source or reference | Identifiers | Additional information |
|---|---|---|---|---|
| Antibody | Anti-DNAH1 (Rabbit polyclonal) | Cusabio | CSB-PA878961LA01HU | IF (1:100) |
| Antibody | Anti-DNAH3 (Rabbit polyclonal) | Cusabio | CSB-PA823461LA01HU | IF (1:100) |
| Antibody | Anti-DNAH3 (Rabbit polyclonal) | Gift from Prof. Yueqiu Tan, Central South University, China. | | WB (1:200) |
| Antibody | Anti-DNAH6 (Rabbit polyclonal) | Proteintech | 18080–1-AP, RRID: AB_2878493 | IF (1:50), WB (1:150) |
| Antibody | Anti-DNAH8 (Rabbit polyclonal) | Atlas | HPA028447, RRID: AB_10599600 | IF (1:200) |
| Antibody | Anti-DNAH17 (Rabbit polyclonal) | Proteintech | 24488–1-AP, RRID: AB_2879568 | IF (1:50) |
| Antibody | Anti-DNAI1 (Rabbit polyclonal) | Proteintech | 12756–1-AP, RRID: AB_10643244 | IF (1:50) |
| Antibody | Anti-DNALI1 (Rabbit polyclonal) | Proteintech | 17601–1-AP, RRID: AB_2095372 | IF (1:50), WB (1:150) |
| Antibody | Anti-TOM20 (Rabbit polyclonal) | Proteintech | 11802–1-AP, RRID: AB_2207530 | IF (1:50) |
| Antibody | Anti-SLC25A4 (Rabbit polyclonal) | Signalway | 32484, RRID: AB_2941094 | IF (1:100) |
| Antibody | Anti-alpha tubulin (Mouse monoclonal) | Abcam | ab7291, RRID: AB_2241126 | IF (1:500) |

*Continued on next page*

*Continued*

| Reagent type (species) or resource | Designation | Source or reference | Identifiers | Additional information |
|---|---|---|---|---|
| Antibody | Anti-alpha tubulin (Rabbit polyclonal) | Proteintech | 11224–1-AP, RRID: AB_2210206 | WB (1:1000) |
| Antibody | Anti-Rabbit IgG, Alexa Fluor 488 (Goat polyclonal) | Invitrogen | A11008, RRID: AB_143165 | IF (1:1000) |
| Antibody | Anti-Mouse IgG, Alexa Fluor 594 (Goat polyclonal) | Invitrogen | A11005, RRID: AB_141372 | IF (1:1000) |
| Antibody | Anti-acetylated alpha tubulin (Mouse monoclonal) | Abcam | ab24610, RRID: AB_448182 | IF (1:500) |
| Antibody | Anti-Mouse IgG, HRP-conjugated (Goat polyclonal) | Proteintech | SA00001-1, RRID: AB_2722565 | WB (1:5000) |
| Antibody | Anti-Rabbit IgG, HRP-conjugated (Goat polyclonal) | Proteintech | SA00001-2, RRID: AB_2722564 | WB (1:5000) |
| Other | Lectin PNA | Invitrogen | L-32460 | IF (1:50) |

## Human subjects

Four unrelated Han Chinese infertile men and their family members were recruited from West China Second University Hospital of Sichuan University and Women and Children's Hospital of Chongqing Medical University. All patients exhibited a normal karyotype (46 XY) without deletion of the azoospermia factor (AZF) region in the Y-chromosome. All of the participants were provided informed consent, and the study was approved by the ethics committee of West China Second University Hospital.

## Genetic analysis

Peripheral blood samples were obtained from the subjects to extract genomic DNA using a DNA purification kit (TIANGEN, DP304). For WES, 1 µg of genomic DNA was utilized for exon capture using the Agilent SureSelect Human All Exon V6 Kit and sequenced on the Illumina HiSeq X system (150 bp read length). The quality of WES, including clean reads, sequencing depth, sequencing coverage, and mapping quality are listed in *Figure 1—source data 1*. The variants identified through WES were annotated and filtered using Exomiser. Next, the variants were screened to obtain candidate variants based on the following criteria: (*Cox et al., 2022*) the allele frequency in the East Asian population was less than 1% in any database, including the ExAC Browser, gnomAD, and the 1000 Genomes Project; (*Eisenberg et al., 2023*) the variants affected coding exons or canonical splice sites; (*Agarwal et al., 2015*) the variants were predicted to be possibly pathogenic or damaging. The remain genes were then analyzed using the Human Protein Atlas (HPA) database (https://www.proteinatlas.org/) and Mouse Genome Informatics (MGI) database (https://informatics.jax.org/) to access their expression in human and mouse testis. Additionally, OMIM database (https://www.omim.org/) and relevant literature were used to understand their relationship with human infertility. Given the assumption of a recessive inheritance pattern, monoallelic variants were excluded from consideration. The remained candidate pathogenic variants were verified by Sanger sequencing on DNA from the patients' families.

## Electron microscopy

For scanning electron microscopy (SEM), sperm samples were fixed in glutaraldehyde (2.5%, w/v) and dehydrated using an ethanol gradient (30, 50, 75, 85, 95, and 100% ethanol). The samples were dried using a $CO_2$ critical-point dryer (Eiko HCP-2, Hitachi) and observed under SEM (S-3400, Hitachi).

For transmission electron microscopy (TEM), sperm samples were fixed in glutaraldehyde (3%, w/v) and osmium tetroxide (1%, w/v) and dehydrated with an ethanol gradient. The samples were embedded in Epon 812. Ultrathin sections were stained with uranyl acetate and lead citrate and analyzed under TEM (Tecnai G2 F20).

## STA-PUT velocity sedimentation

Single testicular cells from obstructive azoospermia and 8-week-old C57BL male mice were obtained using the STA-PUT velocity sedimentation method as described previously (*Liu et al., 2015*; *Chang et al., 2011*). In brief, total spermatogenic cells were harvested by digesting seminiferous tubules with collagenase (Invitrogen, 17100017), trypsin (Sigma, T4799) and DNase (Promega, M6101) for 15 min each at 37 °C. Cells were diluted in bovine serum albumin (BSA, 3%, w/v) and filtered through an 80 mm mesh to remove fragments. Then, the cells were resuspended in BSA (3%, w/v) and loaded into an STA-PUT velocity sedimentation cell separator (ProScience) to obtain germ cells at different stages.

## RNA isolation and quantitative PCR (qPCR)

Total RNA of mouse tissues was extracted using TRIzol reagent (Invitrogen,15596026,) and reverse-transcribed using the 1st Strand cDNA Synthesis Kit (Yeasen, HB210629) according to the manufacturer's instructions. qPCR was carried out on an iCycler RT–PCR Detection System (Bio-Rad Laboratories) using SYBR Green qPCR Master Mix (Bimake, B21202).

## Immunofluorescence staining

Sperm samples were fixed in paraformaldehyde (4%, w/v), permeabilized with Triton X-100 (0.3% v/v) and blocked with BSA (3%, w/v) at room temperature. Samples were incubated with primary antibodies overnight at 4 °C. The next day, the samples were washed and incubated with the secondary antibody, and the nuclei were labeled with 4′,6-diamidino-2-phenylindole (DAPI, Sigma–Aldrich, D9542). Image capture was performed by a laser scanning confocal microscope (Olympus, FV3000).

For staining of mouse tissues, samples were first fixed in paraformaldehyde (4%, w/v) and dehydrated with an ethanol gradient. Then, the samples were embedded in paraffin and sliced into 5 μm sections. After deparaffinization and rehydration, sections were processed with 3% hydrogen peroxide and incubated in sodium citrate for antigen repair. Subsequently, sections were blocked with goat serum and incubated with primary antibodies at 4 °C overnight. The next day, the sections were incubated with the secondary antibody, followed by labeling the nuclei with DAPI. Image capture was performed using a fluorescence microscope (Zeiss, Ax10).

## Western blotting

Sperm samples were lysed in RIPA buffer (Beyotime, P0013B) to extract the total protein. For analysis of DNALI1, the protein samples were mixed with SDS loading buffer (P0015, Beyotime, China), boiled at 95 °C for 5 min, and separated by 12.5% SDS-PAGE. For analysis of DNAH1, DNAH3, DNAH6, the protein samples were mixed with NuPAGE LDS sample buffer (Invitrogen, NP0007), denatured at 70 °C for 10 min, and separated by 3–8% NuPAGE Tris-Acetate gels (EA0375BOX, Invitrogen). Then the resolved proteins were transferred to 0.45 μm PVDF membranes (Merck Millipore, IPVH00010). The membranes were blocked, incubated with primary antibodies at 4°C overnight. The following day, membranes were washed and incubated with HRP-conjugated secondary antibody. Protein bands were visualized using enhanced chemiluminescence reagents (Millipore, WBKLS0500).

## Histology hematoxylin-eosin (H&E) staining

Tissue samples from mice were fixed with 4% paraformaldehyde (w/v) overnight. Following dehydration by ethanol, the samples were embedded in paraffin and sliced into 5 μm sections. The sections were stained with hematoxylin and eosin and observed under a microscope (Zeiss, Axio Imager 2).

## Generation of the *Dnah3* KO mouse model

Animal experiments in this study were approved by the Experimental Animal Management and Ethics Committee of West China Second University Hospital, Sichuan University (No. 20230150), and complied with the Animal Care and Use Committee of Sichuan University. A *Dnah3* knockout mouse model was generated by the CRISPR–Cas9 system. Briefly, Cas9 and signal-guide RNAs (5′-GTATCAAGTGGA

TGTAAACC-3') were transcribed using T7 RNA polymerase in vitro and comicroinjected into the cytoplasm of single-cell C57BL/6 J mouse embryos to generate frameshift mutations by nonhomologous recombination through introduction of a 1 bp insertion in exon 13. Then, the embryos were cultured and transferred into the oviducts of pseudopregnant female mice at 0.5 days post-coitum. A mutation of *Dnah3* in the founder mouse and their offspring was confirmed using PCR and Sanger sequencing.

## Intracytoplasmic sperm injection (ICSI)

ICSI was carried out using standard techniques. In brief, one-month-old female KM mice were injected with 5 IU of equine chorionic gonadotropin (eCG) (ProSpec, HOR-272) to induce superovulation. Metaphase II-arrested (MII) oocytes were acquired through another injection of 5 IU human chorionic gonadotropin after 48 hr. MII oocytes were incubated with Chatot-Ziomek-Bavister medium (Easycheck, M2750) at 37.5 °C and 5% $CO_2$ until use. Mouse cauda epididymal spermatozoa were incubated in human tubal fluid (HTF) medium (Easycheck, M1150) and then frozen and thawed repeatedly to remove sperm tails. For ICSI, a single sperm head was microinjected into an MII oocyte by using a NIKON inverted microscope and a Piezo (PrimeTech, Osaka, Japan) in Whitten's-HEPES medium containing 0.01% polyvinyl alcohol (Gibco,12360–038) and cytochalasin B (3.5 g/ml; Sigma–Aldrich, C-6762). The successfully injected oocytes were transferred into G1-Plus medium (Vitrolife, 10132) and incubated at 37.5 °C and 5% $CO_2$. The animal experiments were approved by the Experimental Animal Management and Ethics Committee of West China Second University Hospital, Sichuan University.

## Statistical analysis

Prism (version 8.4.0, GraphPad, Boston, MA, USA) and SPSS (version 18.0, IBM Corporation, Armonk, NY, USA) were used to perform statistical analyses. All data are presented as the means ± SEMs. Data from two groups were compared using an unpaired, parametric, two-sided Student's *t* test, and a p value less than 0.05 was considered statistically significant.

## Acknowledgements

The authors thank the patients and their family members for their voluntary participation. We are grateful to Guiping Yuan (Analytical and Testing Center of Sichuan University) and Yan Liang (Research Core Facility of West China Hospital, Sichuan University) for their help with TEM images and preparing histology slides. We are grateful to Prof. Yueqiu Tan (Central South University, Changsha, China) for providing the anti-DNAH3 antibody. This work was supported by National Natural Science Foundation of China (82301807).

## Additional information

### Funding

| Funder | Grant reference number | Author |
| --- | --- | --- |
| National Natural Science Foundation of China | 82301807 | Tingting Lin |

The funders had no role in study design, data collection and interpretation, or the decision to submit the work for publication.

### Author contributions

Xiang Wang, Data curation, Formal analysis, Investigation, Methodology, Writing – original draft; Gan Shen, Chuan Jiang, Data curation, Formal analysis; Yihong Yang, Resources, Investigation, Methodology; Tiechao Ruan, Formal analysis, Methodology; Xue Yang, Validation, Methodology; Liangchai Zhuo, Yangdi Ou, Xinya Zhao, Xiangrong Tang, Investigation; Yingteng Zhang, Investigation, Methodology; Shunhua Long, Methodology; Tingting Lin, Conceptualization, Supervision, Funding acquisition, Project administration, Writing – review and editing; Ying Shen, Conceptualization, Supervision, Project administration, Writing – review and editing

## Author ORCIDs
Xiang Wang ⓘ https://orcid.org/0000-0002-8930-6801
Tiechao Ruan ⓘ https://orcid.org/0000-0002-2924-0230
Ying Shen ⓘ https://orcid.org/0000-0002-6346-1002

## Ethics
This study was approved by Ethical Review Board of West China Second University Hospital, Sichuan University (No. 2023160). Informed consent was obtained from each participate in this study before taking part.

The animal experiments were approved by the Experimental Animal Management and Ethics Committee of West China Second University Hospital, Sichuan University (No. 2023150).

Reviewer #1 (Public review): https://doi.org/10.7554/eLife.96755.4.sa1
Reviewer #2 (Public review): https://doi.org/10.7554/eLife.96755.4.sa2
Reviewer #3 (Public review): https://doi.org/10.7554/eLife.96755.4.sa3
Author response https://doi.org/10.7554/eLife.96755.4.sa4

# Additional files

## Supplementary files
MDAR checklist

## Data availability
The published article includes all datasets generated or analyzed during this study. The whole exome-sequencing data were deposited in the NCBI BioProject database under accession PRJNA1270421.

The following dataset was generated:

| Author(s) | Year | Dataset title | Dataset URL | Database and Identifier |
| --- | --- | --- | --- | --- |
| Wang X, Shen G, Yang Y, Jiang C, Ruan T, Yang X, Zhuo L, Zhang Y, Ou Y, Zhao X, Long S, Tang X, Lin T, Shen Y | 2025 | Genetic factors related with infertility | https://www.ncbi.nlm.nih.gov/bioproject/PRJNA1270421 | NCBI BioProject, PRJNA1270421 |

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
