## [Editor Report · eLife Assessment]

This **important** study identifies biallelic variants of DNAH3 in unrelated infertile men and reports infertility in DNAH3 knockout mice. The authors demonstrate that compromised DNAH3 activity decreases the expression of IDA-associated proteins in the spermatozoa of human patients and knockout mice, providing **convincing** evidence that DNAH3 is a novel pathogenic gene for asthenoteratozoospermia and male infertility. The study will be of substantial interest to clinicians, reproductive counselors, embryologists, and basic researchers working on infertility and assisted reproductive technology.

---

## [Referee Report · Reviewer #1 (Public review)]

Summary:

Wang and colleagues identify biallelic variants of DNAH3 in four unrelated Han Chinese infertile men through whole-exome sequencing, which contributes to abnormal sperm flagellar morphology and ultrastructure. To investigate the importance of DNAH3 in male infertility, the authors generated crispant Dnah3 knockout (KO) male mice. They observed that KO mice are also infertile, showing a severe reduction in sperm movement with abnormal IDA (inner dynein arms) and mitochondrion structure. Moreover, nonfunctional DNAH3 expression decreased the expression of IDA-associated proteins in the spermatozoa of patients and KO mice, which are involved in the disruption of sperm motility. Interestingly, the infertility of patients and KO mice is rescued by intracytoplasmic sperm injection (ICSI). Taken together, the authors propose that DNAH3 is a novel pathogenic gene for asthenoterozoospermia and male infertility.

Strengths:

This work investigates the role of DNAH3 in sperm mobility and male infertility. By using gold-standard molecular biology techniques, the authors demonstrate with exquisite resolution the importance of DNAH3 in sperm morphology, showing strong evidence of its role in male infertility. Overall, this is a very interesting, well-written, and appealing article. All aspects of the study design and methods are well described and appropriate to address the main question of the manuscript. The conclusions drawn are consistent with the analyses conducted and supported by the data.

Weaknesses:

The paper is solid, and in its current form, I have not detected relevant weaknesses.

---

## [Referee Report · Reviewer #2 (Public review)]

Wang et al. investigated the role of dynein axonemal heavy chain 3 (DNAH3) in male infertility. They found that variants of DNAH3 were present in four infertile men, and the deficiency of DNAH3 in sperm affects sperm mobility. Additionally, they showed that Dnah3 knockout male mice are infertile. Furthermore, they demonstrated that DNAH3 influences inner dynein arms by regulating several DNAH proteins. Importantly, they showed that intracytoplasmic sperm injection (ICSI) can rescue the infertility in Dnah3 knockout mice and two patients with DNAH3 variants.

Strengths:

The conclusions of this paper are well-supported by data.

Weaknesses:

The sample/patient size is small; however, the findings are consistent with those of a recent study on DNAH3 in male infertility with 432 patients.

---

## [Referee Report · Reviewer #3 (Public review)]

Summary:

(1) To further explore the genetic basis of asthenoteratozoospermia, the authors performed whole-exome sequencing analyses among infertile males affected by asthenoteratozoospermia. Four unrelated Han Chinese patients were found to carry biallelic variations of DNAH3, a gene encoding IDA-associated protein.

(2) To verify the function of IDA associated protein DNAH3, the authors generated a Dnah3-KO mouse model and revealed that the loss of DNAH3 leads to severe male infertility as a result of the severe reduction in sperm movement with the abnormal IDA and mitochondrion structures.

(3) Mechanically, they confirmed decreased expression of IDA-associated proteins (including DNAH1, DNAH6 and DNALI1) in the spermatozoa from patients with DNAH3 mutations and Dnah3-KO male mice.

(4) Then, they also found that male infertility caused by DNAH3 deficiency could be rescued by intracytoplasmic sperm injection (ICSI) treatment in humans and mice.

Strengths:

(1) In addition to existing research, the authors provided novel variants of DNAH3 as important factors leading to asthenoteratozoospermia. This further expands the spectrum of pathogenic variants in asthenoteratozoospermia.

(2) By mechanistic studies, they found that DNAH3 deficiency led to decreased expression of IDA-associated proteins, which may be used to explain the disruption of sperm motility and reduced fertility caused by DNAH3 deficiency.

(3) Then, successful ICSI outcomes were observed in patients with DNAH3 mutations and Dnah3 KO mice, which will provide an important reference for genetic counselling and clinical treatment of male infertility.

---

## [Author Response]

The following is the authors’ response to the previous reviews.

**Public Reviews:**

**Reviewer #1 (Public Review):**
Summary:Wang and colleagues identify biallelic variants of DNAH3 in four unrelated Han Chinese infertile men through whole-exome sequencing, which contributes to abnormal sperm flagellar morphology and ultrastructure. To investigate the importance of DNAH3 in male infertility, the authors generated crispant Dnah3 knockout (KO) male mice. They observed that KO mice are also infertile, showing a severe reduction in sperm movement with abnormal IDA (inner dynein arms) and mitochondrion structure. Moreover, nonfunctional DNAH3 expression decreased the expression of IDA-associated proteins in the spermatozoa of patients and KO mice, which are involved in the disruption of sperm motility. Interestingly, the infertility of patients and KO mice is rescued by intracytoplasmic sperm injection (ICSI). Taken together, the authors propose that DNAH3 is a novel pathogenic gene for asthenoterozoospermia and male infertility.Strengths:This work investigates the role of DNAH3 in sperm mobility and male infertility. By using gold-standard molecular biology techniques, the authors demonstrate with exquisite resolution the importance of DNAH3 in sperm morphology, showing strong evidence of its role in male infertility. Overall, this is a very interesting, well-written, and appealing article. All aspects of the study design and methods are well described and appropriate to address the main question of the manuscript. The conclusions drawn are consistent with the analyses conducted and supported by the data.Weaknesses:The paper is solid, and in its current form, I have not detected relevant weaknesses.

We thank the comments from the reviewer very much.

**Reviewer #2 (Public Review):**
Wang et al. investigated the role of dynein axonemal heavy chain 3 (DNAH3) in male infertility. They found that variants of DNAH3 were present in four infertile men, and the deficiency of DNAH3 in sperm affects sperm mobility. Additionally, they showed that Dnah3 knockout male mice are infertile. Furthermore, they demonstrated that DNAH3 influences inner dynein arms by regulating several DNAH proteins. Importantly, they showed that intracytoplasmic sperm injection (ICSI) can rescue the infertility in Dnah3 knockout mice and two patients with DNAH3 variants.Strengths:The conclusions of this paper are well-supported by data.Weaknesses:The sample/patient size is small; however, the findings are consistent with those of a recent study on DNAH3 in male infertility involving 432 patients.

We extend our sincere gratitude to the expert reviewers for their valuable comments and insightful suggestions.

A cohort of 587 unrelated infertile men with asthenoteratozoospermia was recruited to investigate the potential genetic etiology using WES. In addition to mutations in *DNAH3* identified in four patients, mutations in serval other genes previous reported by our group, including *CFAP65* (Zhang et al., 2019. PMID: 31571197), (Yang et al., 2020. PMID: 32681648), (Li et al., 2022. PMID: 34791246), (Zheng et al., 2023. PMID: 35654582), (Zhang et al., 2022. PMID: 35296684), (Zhang et al., 2022. PMID: 36206347), (Zhang et al., 2023. PMID: 36481789), (Wang et al., 2023. PMID: 36415156), (Zhang et al., 2024. PMID: 38228861), (Jin et al., 2023. PMID: 38126872), (Ruan et al., 2023. PMID: 36967801), (Liu et al., 2022. PMID: 35821214), as well as other unreported variants were also identified.

**Reviewer #3 (Public Review):**
Summary:(1) To further explore the genetic basis of asthenoteratozoospermia, the authors performed whole-exome sequencing analyses among infertile males affected by asthenoteratozoospermia. Four unrelated Han Chinese patients were found to carry biallelic variations of DNAH3, a gene encoding IDA-associated protein.(2) To verify the function of IDA associated protein DNAH3, the authors generated a Dnah3-KO mouse model and revealed that the loss of DNAH3 leads to severe male infertility as a result of the severe reduction in sperm movement with the abnormal IDA and mitochondrion structures.(3) Mechanically, they confirmed decreased expression of IDA-associated proteins (including DNAH1, DNAH6 and DNALI1) in the spermatozoa from patients with DNAH3 mutations and Dnah3-KO male mice.(4) Then, they also found that male infertility caused by DNAH3 deficiency could be rescued by intracytoplasmic sperm injection (ICSI) treatment in humans and mice.Strengths:(1) In addition to existing research, the authors provided novel variants of DNAH3 as important factors leading to asthenoteratozoospermia. This further expands the spectrum of pathogenic variants in asthenoteratozoospermia.(2) By mechanistic studies, they found that DNAH3 deficiency led to decreased expression of IDA-associated proteins, which may be used to explain the disruption of sperm motility and reduced fertility caused by DNAH3 deficiency.(3) Then, successful ICSI outcomes were observed in patients with DNAH3 mutations and Dnah3 KO mice, which will provide an important reference for genetic counselling and clinical treatment of male infertility.

We are very grateful for the reviewer's careful comments.

**Recommendations for the authors:**

**Reviewer #1 (Recommendations for The Authors):**
I have carefully read the revised versions of this manuscript, and I would like to thank the authors for addressing all my previous concerns.I have no additional comments or suggestions.

We thank the reviewer for reviewing our revised manuscript.

**Reviewer #2 (Recommendations for The Authors):**
(1) Statistical analyses should be provided alongside the quantification (Fig S1B, S7C).

According to the suggestions of the reviewer, we have added statistical analyses of the corresponding quantification in the legends of Figure S1 and Figure S7.

(2) The numbers of sperms counted in Fig S1A should be listed.

In response to reviewer's valuable suggestions. We have listed the corresponding ratio of different morphological defects in sperm tail of the patients in Figure S1A.

(3) Due to the high similarities in experimental design, data and conclusions between the current study and previously published work by Meng et al. (2024), as well as the very similar titles of the two studies, it is crucial to emphasize the differences in the Discussion section.

Many thanks for reviewer's kind suggestions for our revised manuscript.

Employing whole-exome sequencing (WES) on infertile men to identify candidate variants, followed by *in-silico* and functional analysis of these variants, and generating mouse models using CRISPR-Cas9 technology, has proven to be an efficient and widely used approach for uncovering the causative genes of male infertility associated with sperm defects. Both our study and the recent work by Meng et al. utilized this approach to verify whether DNAH3 mutations are a cause of asthenoteratozoospermia. Additionally, we have also updated the title of our study to: 'DNAH3 deficiency causes flagellar inner dynein arm loss and male infertility in humans and mice'.

Meng et al. reported *DNAH3* mutations in asthenoteratozoospermia affected patients, revealing multiple morphological defects in sperm tail. Moreover, ultrastructural abnormalities of the flagellar axoneme in the patients were evident in these patients, characterized by a disrupted '9+2' arrangement and the notable absence of IDAs. Additionally, they generated *Dnah3* KO mice, which were infertile and exhibited moderate morphological abnormalities. While the '9+2' microtubule arrangement in the flagella of their *Dnah3* KO mice remained intact, the IDAs on the microtubules were partially absent. In our study, we observed similar phenotypic differences between *DNAH3*-deficient patients and *Dnah3* KO mice. Both studies suggest that DNAH3 plays a crucial role in human and mouse male reproduction.

However, there are notable differences between the two studies. Firstly, the phenotypes of *Dnah3* KO mice showed slight differences. Meng et al. generated two *Dnah3* KO mouse models (KO1 and KO2), and both of which exhibited significantly higher sperm motility and progressive motility than in our study, where nearly all sperm were completely immobile. Furthermore, their *Dnah3* KO2 mice even displayed motility comparable to WT mice and retained partial fertility. We speculate that these differences may be attributed to variations in mouse genetic background or the presence of a truncated DNAH3 protein resulting from specific knockout strategies. Secondly, we conducted additional research and uncovered novel findings. We revealed that male infertility caused by *DNAH3* mutations follows an autosomal recessive inheritance pattern, as confirmed through Sanger sequencing of the patients' parents. We also discovered the dynamic expression and localization of DNAH3 during spermatogenesis in humans and mice through immunofluorescent staining. We further found that DNAH3 deficiency had no impact on ciliary development in the oviduct or on oogenesis in mice, resulting in normal female fertility. Moreover, in the absence of DNAH3 in both humans and mice, the expression of IDA-associated proteins, including DNAH1, DNAH6 and DNALI1, was decreased, while the expression of ODA-associated proteins remained unaffected, indicating that DNAH3 is involved in sperm axonemal development, specifically through its role in the assembly of IDAs. Collectively, our study corroborates the findings of Meng et al., and provides additional unique insights, comprehensively elucidating the critical role of DNAH3 in human and mouse spermatogenesis.

We have added these discussions in line 275 to line 306.

**Reviewer #3 (Recommendations for The Authors):**
I have no more recommendations for the authors.

We thank the reviewer for reviewing our revised manuscript.